# Visualization of Chromatin in the Yeast Nucleus and Nucleolus Using Hyperosmotic Shock

**DOI:** 10.3390/ijms22031132

**Published:** 2021-01-24

**Authors:** Nicolas Thelen, Jean Defourny, Denis L. J. Lafontaine, Marc Thiry

**Affiliations:** 1Unit of Cell Biology, GIGA-Neurosciences, CHU Sart-Tilman, University of Liège, B36, 4000 Liège, Belgium; nthelen@uliege.be (N.T.); jean.defourny@uliege.be (J.D.); 2RNA Molecular Biology, Fonds de la Recherche Scientifique (F.R.S./FNRS), Center for Microscopy and Molecular Imaging (CMMI), Université libre de Bruxelles (ULB), B-6041 Gosselies, Belgium; denis.lafontaine@ulb.ac.be

**Keywords:** yeast, nucleus, chromatin, nucleolus, perichromatin granules, electron cytochemistry, immunoelectron microscopy

## Abstract

Unlike in most eukaryotic cells, the genetic information of budding yeast in the exponential growth phase is only present in the form of decondensed chromatin, a configuration that does not allow its visualization in cell nuclei conventionally prepared for transmission electron microscopy. In this work, we studied the distribution of chromatin and its relationships to the nucleolus using different cytochemical and immunocytological approaches applied to yeast cells subjected to hyperosmotic shock. Our results show that osmotic shock induces the formation of heterochromatin patches in the nucleoplasm and intranucleolar regions of the yeast nucleus. In the nucleolus, we further revealed the presence of osmotic shock-resistant DNA in the fibrillar cords which, in places, take on a pinnate appearance reminiscent of ribosomal genes in active transcription as observed after molecular spreading (“Christmas trees”). We also identified chromatin-associated granules whose size, composition and behaviour after osmotic shock are reminiscent of that of mammalian perichromatin granules. Altogether, these data reveal that it is possible to visualize heterochromatin in yeast and suggest that the yeast nucleus displays a less-effective compartmentalized organization than that of mammals.

## 1. Introduction

The eukaryotic cell is characterised by the presence of a compartment bounded by a double membrane, called the nucleus, which contains genetic information. The latter is organised into chromatin domains with distinct structure and function [1,2]. Heterochromatin is usually condensed and lacks highly expressed genes [3]. It concentrates mainly at the periphery of the nucleus and around the nucleolus. Euchromatin is usually decondensed allowing gene expression. It is mainly located at the periphery of the heterochromatin clumps, a region of the nucleoplasm called the perichromatin spaces [4]. In addition to chromatin, the nucleus also contains various membraneless subcompartments formed by liquid–liquid phase separation (LLPS), which concentrates specific proteins and excludes others to form RNA and protein-rich microenvironments that promote or inhibit certain activities. These are now referred to as biomolecular condensates. The best-known nuclear subcompartment is the nucleolus, the assembly site of ribonucleoprotein particles, primarily dedicated to ribosome biogenesis [5,6,7]. Recent data indeed support the notion that the nucleolus is a multilayered biomolecular condensate whose formation by LLPS facilitates the initiation of the first stages of ribosome biogenesis and other functions [6].

The yeast *Saccharomyces cerevisiae*, an extensively used experimental model, displays a small nucleus of ~2 µm in diameter in phase G1 of the cell cycle. In haploid yeast strains, this nucleus contains a genome of ~13 Mb organised into 16 chromosomes, a genome-to-nucleus size ratio very similar to that of human cells [8]. The nucleolus represents its most visible subcompartment, as it occupies one third of its volume. The rest of the nucleus does not seem to contain other structural subcompartments as those seen in metazoans, but it is rich in DNA. Curiously, the yeast nucleus does not show condensed chromatin on the ultrathin sections classically prepared for transmission electron microscopy [9,10,11,12]. In addition, electron cryotomography studies confirmed that yeast chromatin did not form a compact structure in interphase and mitosis [13,14]. Altogether, these studies suggest that in yeast all the chromatin is present in an “open” configuration, which would enable high transcription rates. This peculiarity of the yeast chromatin makes it difficult to establish its spatial relationships to the nucleolus.

In this work, we studied the distribution of chromatin in the yeast nucleus at the ultrastructural level and its relationships to the nucleolus using different cytochemical and immunocytological techniques. We applied these approaches to cells subjected or not to hyperosmotic shock, a treatment known to induce chromatin condensation in eukaryotic cells [15,16]. Our results show that most of the yeast chromatin condenses in the form of electron-dense clumps when the cells are subjected to hyperosmotic shock. This chromatin is mainly detected in the nucleoplasmic region, but a small part is also detected in the nucleolar region. A few clumps are associated with the nuclear envelope. We also revealed osmotic shock resistant DNA in the fibrillar strands of the nucleolus. The latter take on a pinnate appearance reminiscent of ribosomal genes undergoing active transcription in the form of Christmas trees observed after molecular spreading. We also showed an increase in the number of perichromatin grains upon hyperosmotic treatment and reveal these to be always associated with chromatin as is the case in mammalian cells. 

## 2. Results

To determine the distribution of chromatin in the nucleus of exponentially growing *Saccharomyces cerevisiae* cells, we first applied the cytochemical method of acetylation at the ultrastructural level. This method is known to improve the contrast of the condensed chromatin within the cell nucleus [17], as formerly shown in another yeast, *Candida albicans* [18]. It also makes it possible to clearly distinguish the different constituents of the nucleolus [19].

Under these experimental conditions, we did not observe condensed chromatin in the yeast nucleus (Figure 1A,B). The latter appears as two distinct morphological zones, corresponding, respectively, to a large nucleolus (circled by a dotted line) and the extranucleolar space or nucleoplasm. The nucleolus is attached to the nuclear envelope and essentially comprises two distinct compartments: strands of material consisting of fine fibrils (F), which are surrounded by granular clusters (G). In some nucleoli, we also detected, on occasion, areas with a texture and electron density similar to that of the nucleoplasm (Figure 1A, asterisk). These heterogeneous areas were sometimes located in contact with the nuclear envelope. The nucleoplasm does not contain interchromatin granule clusters or nuclear bodies, such as those classically observed in the nucleus of mammalian cells. Instead, it contains numerous scattered granules, which are similar in size to nucleolar granules (Figure 1B, arrowheads and Table 1). In addition, in some ultrathin sections (12%, *n* = 51), one or a few large nucleoplasmic granules are observed (Figure 1B, large arrow). These granules are significantly larger than the many nucleoplasmic or nucleolar granules (Table 1).

To accurately determine the location of chromatin in the yeast nucleus, we then applied the immunocytological technique of DNA detection by terminal deoxyribonucleotide transfer on acetylated yeast sections. The labelling is mainly observed in the nucleoplasmic area, as expected, since it is known as the DNA-rich area (Figure 1C), although fewer gold particles are also seen in the nucleolus, which likely correspond to the rDNA units. In the nucleoplasmic zone, the labelling is dispersed rather homogenously. In the nucleolus, labelling is mainly found at a few heterogeneous areas and a few gold particles are also detected on the fibrillar cords. A quantitative analysis clearly confirms our qualitative assessment (Table 2): a significant labelling is observed at the nucleoplasmic zone, as well as in the heterogeneous areas and fibrillar cords of the nucleoli.

To better identify the different nucleolar subcompartments, we then applied a cytochemical staining for the detection of argyrophilic proteins of the nucleolar organizer. An electron opaque silver precipitate is deposited essentially on the fibrillar strands of the nucleolus (Figure 1D). In contrast, the granular zone and the heterogeneous nucleolar areas are conspicuously devoid of precipitates.

Since hyperosmotic treatment is known to induce chromatin condensation in eukaryotic cells of different origins [15,16], we reasoned that applying such a treatment to yeast cells might render its chromatin visible. To visualize the chromatin in the yeast nucleus after hyperosmotic treatment, we applied our cytochemical acetylation method to cells incubated in a medium containing 1M sorbitol prior to fixation. Remarkably, such hyperosmotic shock induced the appearance of electron-dense masses on the ultrathin sections performed on the yeast nucleus (Figure 2A,B, Ch). These masses are mainly observed in the nucleoplasmic zone, sometimes in close association with the nuclear envelope. Some masses are also present in the nucleolus at the previously described heterogeneous areas (Figure 2A, asterisk). These intranucleolar masses of chromatin are frequently connected to the nucleolar fibrillar cords by thin filaments (see Figure 2C). Close examination of the fibrillar cords further reveals the presence of pinnate structures in places, with a central axis connected to lateral branches (Figure 2C,D, arrows). Such structures are somehow reminiscent of actively transcribed ribosomal genes in the so-called “Christmas trees” typically revealed after molecular spreading [20]. The systematic alignment of the pinnate structures is particularly striking.

We also note that dense masses of chromatin are also perfectly visible in the nucleus of cells undergoing division (Figure 2E).

To verify that the detected masses do indeed correspond to chromatin, we next applied the immunocytological technique of DNA detection by TdT on sections of cells placed in a hyperosmotic medium prior to fixation (Figure 2F). Under these experimental conditions, the DNA labelling superimposes nearly perfectly with the nucleoplasmic and intranucleolar electron-dense masses (Figure 2F). A few gold particles are also seen on the fibrillar strands of the nucleoli. A quantitative analysis of the labelling reveals a significant difference in the dense masses present in the nucleoplasmic and nucleolar zones but also in the fibrillar cords of the nucleolus (Table 2).

It is also interesting to note that in addition to the appearance of clumps of condensed chromatin, incubation in hyperosmotic medium also induces an increase in the number of large extranucleolar granules. In total, 78% of the ultrathin sections passing through the yeast nucleus (*n* = 54) contain one or more (up to 7 per section) extranucleolar large granules (see Discussion).

Finally, to establish which of the structures detected in yeast cell nuclei that underwent hyperosmotic shock consist of DNA and which are made of RNA, acetylated sections were submitted to Bernhard’s regressive staining (Figure 3A,B). In this procedure, upon incubation with the chelating agent EDTA, the uranyl stain is removed from DNA preferentially while it remains bound to ribonucleoprotein (RNP). The dense patches of chromatin present in the nucleoplasm (Figure 3A,B, Ch) and in the heterogenous nucleolar areas (Figure 3A, asterisk) lose their staining and appear as white areas (“negative print” aspect), confirming they consist mostly of DNA. In contrast, the two nucleolar compartments, as well as the small and large nucleoplasmic granules, maintain their staining, indicating that they consist mostly of RNP particles. This also reveals that the large nucleoplasmic granules are always associated with the condensed chromatin patches (Figure 3B, arrows).

## 3. Discussion

### 3.1. Chromatin Does Not Form a Condensed Structure in Exponentially Growing Yeast Cells

In exponentially growing yeast cells conventionally prepared for transmission electron microscopy, we showed that the vast majority of nuclear chromatin does not form a condensed structure. This is consistent with previous reports [9,10,11,12]. Even when we used the cytochemical method of acetylation, which is known to increase the contrast of the condensed chromatin [17], we were not able to visualize method of acetylation, which is known to increase the contrast of the condensed chromatin [17], we were not able to visualize heterochromatin in the yeast nucleus. DNA is of course present in the yeast nucleus, and this was revealed at the ultrastructural level by the TdT immunocytological technique (Figure 1C). This DNA is mainly localized in the nucleoplasmic area, as often reported in light microscopy using DAPI staining [21]. We also detected some labelling in the nucleolar region, in particular in heterogeneous areas, and on fibrillar strands.

Our observations seem to clearly indicate that the exponentially growing budding yeast chromatin is essentially decondensed, making it very poorly visible in ultrathin sections. This result is consistent with the fact that, unlike in many eukaryotes, the vast majority of budding yeast chromatin in vivo would not organize into 30 nm fibres, nor into any higher order periodic structure [14,22]. Such an “open” configuration would afford high transcriptional rates, which would seem particularly important for compact gene-rich intron-poor genomes, such as the yeast genome [23].

### 3.2. Budding Yeast Chromatin Forms Condensed Structures upon Hyperosmotic Treatment

We revealed formation of heterochromatin patches both in the nucleoplasm and, to a lesser extent, the nucleolus of yeast cells incubated in a hyperosmotic medium prior to fixation (Figure 2A,B). This observation is consistent with recent in vitro studies. For example, by electron cryotomography, Cai and colleagues indicated that yeast chromatin could condense in vitro into large masses even in the absence of bivalent cations [13]. It was also shown in vitro that it was possible to create compact heterochromatin fibres in the presence of the silent regulatory factors Sir2, Sir3 and Sir4, which form the SIR complex [24]. In addition, the presence of heterochromatin has already been described in the yeast nucleus in a steady state [25]. It has also been visualized during hyperosmotic stress leading to cell death [26]. However, in our case, the treatment duration and sorbitol concentration used were much lower, preserving cell growth. Finally, the condensed chromatin in the exponentially growing budding yeast nucleus was also revealed by treating fixed cells with RNase [9]. However, under these experimental conditions, the other nuclear compartments, such as the nucleolus, were no longer visible, and it was, therefore, not possible to establish the relationships between the chromatin and the other nuclear compartments.

Our approach makes it possible to visualize the chromatin while preserving the other nuclear subcompartments. We found that the heterochromatin is essentially localized in the nucleoplasmic region of the yeast nucleus. We also observed part of the condensed chromatin in the nucleolus, which should correspond to chromosome XII carrying the repeat arrays of ribosomal DNA genes [27]. Some regions of the heterochromatin are also seen in contact with the nuclear envelope; they correspond to the telomeric parts of one or more chromosomes that concentrate the SIR complex [8].

### 3.3. Two Types of DNA Are Present in the Yeast Nucleolus

In mammalian cells, the nucleolus is classically described as three embedded layers, starting with the fibrillar centre (FC) at the inner core of the organelle, surrounded by a dense fibrillar component (DFC), and with the FC/DFC forming a module present in one or multiple copies in a large mass of granules termed the granular component (GC) [28]. Recent analyses have confirmed that RNA polymerase I and upstream binding transcription factor, a co-regulator of RNA polymerase I, are enriched in the cortical area of the FC and that transcriptionally active rRNA genes lie at the interface between the FC and the DFC [29,30]. Loss of nucleolar function is widely known to be associated with the segregation of the three nucleolar components which are no longer embedded within each other but become spatially juxtaposed next to one another [28].

It was proposed that such an organization in three nucleolar subcompartments is a recent evolutionary acquisition that emerged only after the transition between anamniote and amniote organisms [7,19,31]. According to this model, the yeast nucleolus displays only two subcompartments, fibrillar strands (F) and granules (G), with the strands corresponding to sites of active transcription of ribosomal DNA genes, and the granules to maturing precursor ribosomal particles. In yeast, loss of nucleolar function was also shown to lead to segregation of components, but in this case, consistently, only two areas were detected [28].

We revealed two types of DNA associated with the nucleolus: one whose compaction depends upon an osmotic shock to become detectable as patches of heterochromatin in the heterogenous nucleolar areas (Figure 2A), and another visible even when cells are grown under physiological conditions and corresponding to “fibrillar strands” (Figure 1C).

Unlike the FCs of mammalian cells, the yeast heterogenous nucleolar areas with which a patch of heterochromatin is associated upon hyperosmotic shock do not contain RNA polymerase I or any of its associated factors [32,33]. They are also negative for AgNOR staining (Figure 1D) unlike mammalian FCs [34]. In addition, mammalian FC DNA does not condense upon hyperosmotic treatment [15]. For these reasons, the yeast heterogenous nucleolar areas should be considered simply as an equivalent to the so-called mammalian interstices, as discussed previously [7].

The DNA in the nucleolar fibrillar strands, on the other hand, did not appear to be sensitive to hyperosmotic shock. We suggest that this is because they contain actively transcribed rDNA units, which are possibly not prone to being compacted upon this treatment. Consistent with active transcription, we found that fibrillar cords sometimes display a pinnate appearance reminiscent of the “Christmas tree” images of actively transcribed ribosomal genes, as classically revealed by molecular spreading [20].

### 3.4. The Yeast Nucleus Has a Limited Number of Structural Subcompartments

Our ultrastructural examination of the yeast nucleus clearly indicates that it contains only very few structural subcompartments, namely, a nucleolus, which occupies a large nuclear volume, and chromatin, which is not obvious to visualize. However, it does not include subcompartments that are classically described in mammalian cells, such as interchromatin granule clusters or nuclear bodies, e.g., Cajal bodies.

A peculiarity of the yeast nucleus is the presence of numerous granules scattered throughout the nucleoplasmic area. Their size is remarkably close to the size of nucleolar granules, suggesting they are maturing pre-ribosomal particles. This is consistent with the former detection of similarly sized particles by in situ hybridization using 18S or 25S ribosomal RNA probes on ultrathin sections [35]. Alternatively, they could correspond, at least in part, to dispersed interchromatin granules whose diameter in mammalian cells is similar. Immunofluorescence microscopy studies have shown that splicing factors occupy discrete areas in the nucleoplasmic area of the yeast nucleus [36]. However, interchromatin granules in mammalian cells are interconnected structures [37], which does not seem to be the case in budding yeast.

Another surprising observation is the increase in the number of the large granules after hyperosmotic shock. These large granules could represent perichromatin granules, such as those detected in mammals [38]. Indeed, as in mammals, they are associated with condensed chromatin, display a similar size, and consist mostly of RNPs. Finally, the number of mammalian perichromatin granules also increases after various treatments, such as temperature elevation [38].

In conclusion, our detailed ultrastructural observation of the yeast cell nucleus revealed that it is possible to detect heterochromatin both in the nucleoplasm and in the nucleolus under specific treatment (hyperosmotic shock), and that the yeast nucleus contains less numerous subcompartments than that of mammalian cells. The structural compartmentalization of the nucleus likely became more complex during eukaryotic evolution, akin to the result we reported previously for the nucleolus which acquired a third subcompartment at the transition between anamiotic and amniotic vertebrates in the group Reptilia [31].

## 4. Materials and Methods

### 4.1. Biological Material

Standard *Saccharomyces cerevisiae* growth and handling techniques were employed. The wild-type strain used in this study was YDL401 [39].

The hyperosmotic shock treatment was performed as follows: exponentially growing yeast cells at an OD_600_ between 0.2–0.3 were collected, the cell pellet was washed once in 20 mL sterile water and immediately resuspended at room temperature in 20 mL fresh medium comprising 19 mL of 1M sorbitol solution containing 2.5 mM ethylene diamine tetraacetic acid (EDTA) and 1 mL of 1M dithiothreitol. Finally, cells were rinsed once in 20 mL of 1M sorbitol before being fixed. After each wash step, cells were harvested by centrifugation at 1500× *g* for 5 min.

### 4.2. Electron Microscopy 

Cells were fixed for 60 min at 4 °C in 1.6% glutaraldehyde in 0.1 M Sorensen’s buffer (pH 7.4) and acetylated as previously described [40] or stained with the AgNOR technique [41]. After washing in Sorensen’s buffer, cells were dehydrated through graded ethanol solutions and then processed for embedding in epon. Ultrathin sections were mounted on colloidin-coated grids and stained with uranyl acetate and lead citrate before examination in a JEM 1400 transmission electron microscope at 80 kV. 

Bernhard’s EDTA regressive staining [42]: Thin sections of acetylated cells were incubated for 5 min at room temperature in darkness on drops of 50% ethanolic uranyl acetate, rinsed in three 25 mL-beakers filled with boiled deionised water, floated on drops of 0.2 M EDTA (pH 7) for 10–60 min, transferred on drops of aqueous lead citrate for 5 min, rinsed in three 25 mL-beakers filled with boiled deionised water and dried on filter paper.

Detection of DNA: In situ terminal deoxynucleotidyl transferase (TdT)-immunogold method [43].

Acetylated ultrathin sections were incubated for 30 min at 37 °C on the surface of the following medium: 100 mM sodium cacodylate (pH 6.5); 10 mM beta-mercaptoethanol; 2 M MnCl2; 50 µg/mL bovine serum albumin (BSA) (Sigma, St Louis, MO, USA); 20 µM 5-bromo-2-deoxyuridine (BUdR) triphosphate (Sigma); 4 µM each of dCTP, dGTP and dATP (Boehringer Mannheim, Germany); and 125 U/mL TdT (Boehringer; Mannheim, Germany). After three rinses in double-distilled water, the different sections were incubated for 20 min in PBS (0.14 M NaCl, 6 mM Na_2_HPO_4_, 4 mM KH_2_PO_4_, 1% BSA, pH 7.2) containing normal goat serum (NGS) diluted 1:30. After rinses in PBS containing 0.2% BSA, pH 7.2, the next step of the treatment was a 4 h incubation at room temperature with mouse anti-BUdR antibodies (Roche Diagnostics, Indianapolis, IL, USA) diluted 1:50 in PBS pH 7.2 containing 0.2% BSA and 0.2% NGS. After four rinses in PBS containing 1% BSA, pH 7.2 and one in PBS containing 0.2% BSA, pH 8.2, sections were transferred to an incubation medium containing rabbit anti-mouse IgG coupled to colloidal gold (10 nm diameter, Amersham BioSciences, Buckinghamshire, UK) diluted 1:40 in PBS (with 0.2% BSA), pH 8.2, and incubated for 1 h at room temperature. Samples were rinsed with PBS containing 1% BSA, pH 8.2 four times, then four times with distilled water. As controls, TdT or labelled nucleotides were omitted from the TdT medium.

## Figures and Tables

**Figure 1 ijms-22-01132-f001:**
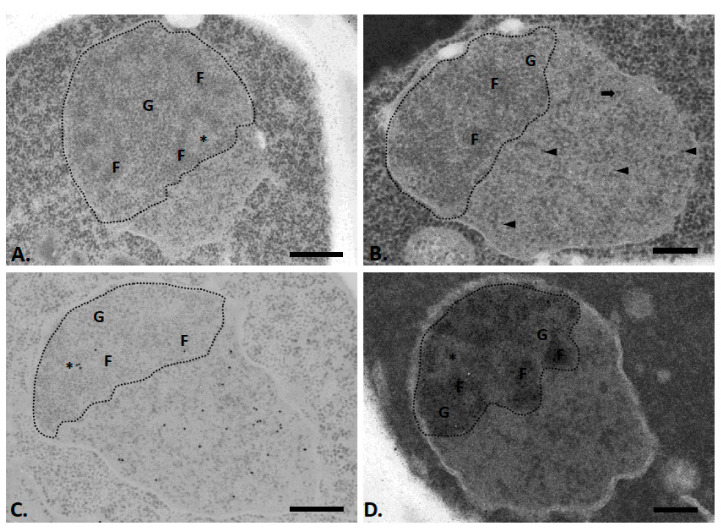
Nuclei of exponentially growing yeast cells, as revealed after the application of the cytochemical method of acetylation (**A**,**B**) combined with the immunocytological technique of detecting the DNA by the transferase of terminal deoxyribonucleotides (**C**) or with the staining of the argyrophilic proteins of the nucleolar organizer (**D**). The nucleolar region is delimited by a dotted line. It contains fibrillar strands (F) surrounded by clusters of granules (G). In addition, heterogeneous areas (asterisks) can be observed within nucleoli. The nucleoplasmic region contains numerous small granules (arrowheads) and rare large granules (thick arrow). Bars = 0.5 µm (**A**,**C**) or 0.25 (**B**,**D**).

**Figure 2 ijms-22-01132-f002:**
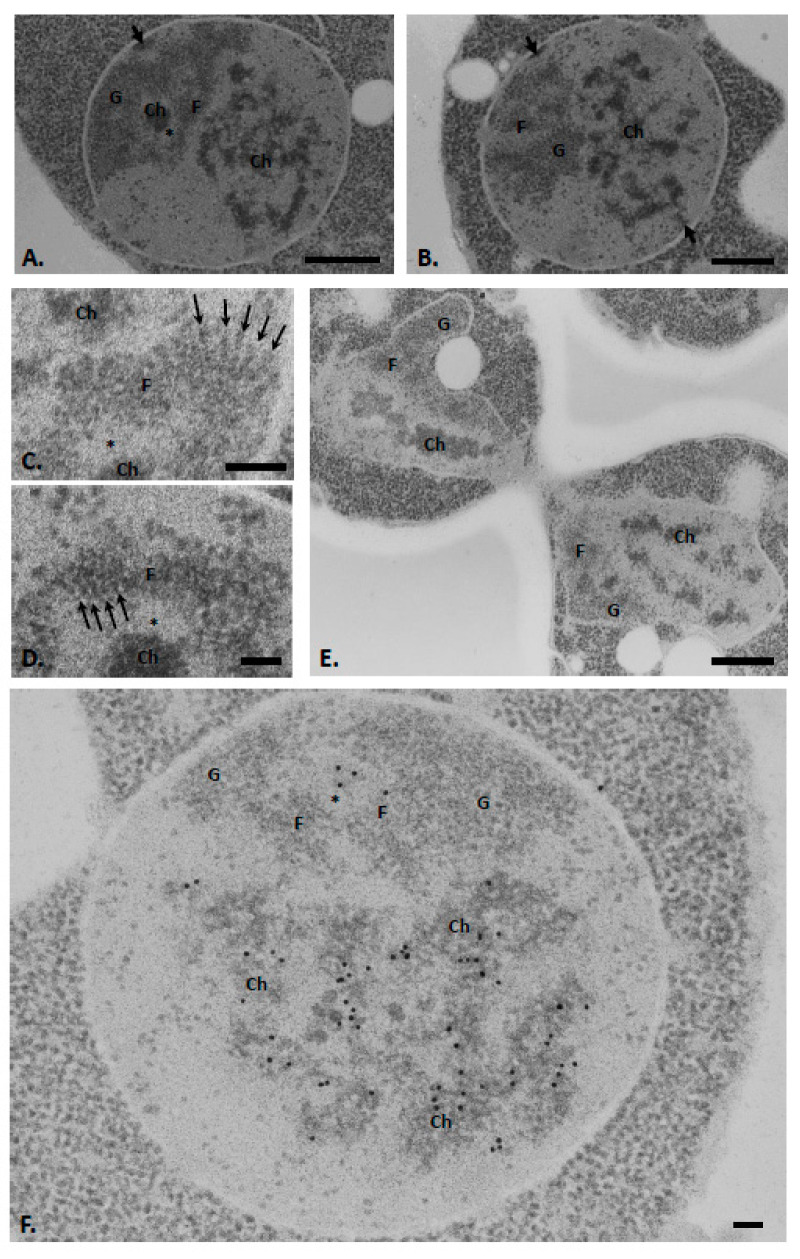
Nuclei of exponentially growing yeast cells exposed to hyperosmotic shock as revealed after the application of the cytochemical method of acetylation (**A**–**E**) combined with the immunological technique of DNA detection by the transferase of terminal deoxyribonucleotides (**F**). (**A**,**B**) Dense masses (Ch, chromatin) are mainly observed in the nucleoplasmic region. Heterogeneous areas (asterisks) in the nucleolar region also contain patches of condensed chromatin. Some dense masses conspicuously associated with the nuclear envelope (thick arrows). (**C**,**D**) Some portions of the fibrillar strands (**F**) of the nucleolus have a pennate appearance with remarkable alignment (thin arrows). (**E**) Cell in division showing the presence of dense masses of compacted chromatin in the two forming nuclei. (**F**) Gold particles labelling DNA are observed essentially on the nucleoplasmic and intranucleolar dense masses. F, fibrillar strands; G, granular zone. Bars = 0.5 (**A**,**B**,**E**) or 0.1 µm (**C**,**D**,**F**).

**Figure 3 ijms-22-01132-f003:**
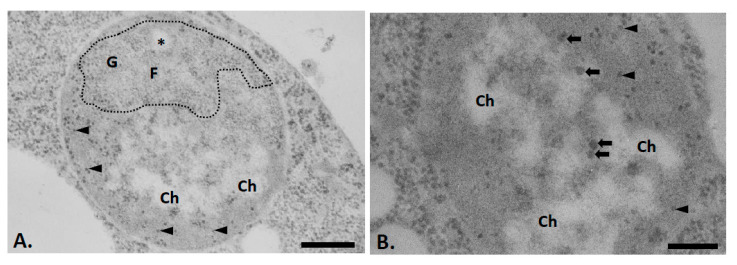
Nuclei of exponentially growing yeast cells subjected to hyperosmotic shock as revealed after the application of the cytochemical method of acetylation combined with the technique of regressive EDTA staining to visualize RNP-containing structures. Dense masses (Ch) present in nucleoplasmic region and in the heterogenous areas of the nucleolus (asterisks) are “bleached” indicating they consisted mostly of DNA. The fibrillar strands (F) and granular masses (G) of the nucleoli remain contrasted as well as the small (arrowheads) and large extranucleolar granules (thick arrows). The latter are always associated with dense masses. Bars = 0.5 (**A**) or 0.25 µm (**B**).

**Table 1 ijms-22-01132-t001:** Mean diameter of granules observed in different yeast compartments and their standard deviation. Student’s *t*-test for different nuclear granules vs. cytoplasmic granules (* *p* < 0.01).

**Cytoplasmic Ribosomes**	21.96 ± 2.38 nm	*n* = 29
**Nucleolar granules**	21.96 ± 3.09 nm	*n* = 27
**Nuclear granules dispersed in nucleoplasmic zone**	20.28 ± 1.99 nm	*n* = 28
**Large nuclear granules in nucleoplasmic zone**	38.52 ± 4.78 nm *	*n* = 35

**Table 2 ijms-22-01132-t002:** Mean density of the label (number of gold particles per square micrometre) obtained in different compartments of yeast cells with or without passage in 1M sorbitol solution, as revealed after the application of the immunocytological technique using TdT as well as the standard deviation. Student’s *t*-test for nuclear compartments vs. resin or cytoplasm (* *p* < 0.01).

Cell Compartments	Cells without Passage in 1M Sorbitol Solution *n* = 11	Cells with Passage in 1M Sorbitol Solution *n* = 13
NUCLEOLUS		
Fibrillar cordons	3.68 ± 2.55 *	9.48 ± 5.40 *
Granular component	0	0.20 ± 0.72
Pale areas	65.73 ± 23.65 *	79.29 ± 34.80 *
NUCLEOPLASMIC ZONE	31.86 ± 10.49 *	28.98 ± 7.70 *
Condensed chromatin	/	70.20 ± 18.29 *
Interchromatin area	/	1.07 ± 1.20
CYTOPLASM	0.24 ± 0.31	1.53 ± 0.94
RESIN	0.24 ± 0.37	1.69 ± 1.50

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
