# Peer review of "Visualization of Chromatin in the Yeast Nucleus and Nucleolus Using Hyperosmotic Shock"

_ijms, 2021, doi:10.3390/ijms22031132_

Round 1
Reviewer 1 Report
The paper by Thelen and coworkers studies at EM the chromatin distribution in yeast both in control condition and after hyperosmotic shock. The rationale behind is to induce chromatin condensation in a model normally devoid of heterochromatin areas. The chromatin clumps are detected mainly in the nucleoplasm and at the nucleolar level. Interestingly, an increase in PG-like structures occurs.
The paper is sound, well written and the discussion well supported by the results.
I have but a couple or remarks:
- in the original paper, Bernhard stated that the EDTA regressive technique is preferential for RNPs toward DNPs, but never claimed it to be specific for RNA alone. Since it is clear enough that there is a clear segregation in the nucleus, I suggest to modify RNA into RNPs.
- Tchalidze et al. (2019) is quoted in the text but not in the references
- as a suggestion for further work, the sections could be stained with osmium ammine. Given the very high resolution of the technique, this could clarify where DNA is inside the pinnate structures seen.
Author Response
We corrected this sentence by changing the RNA to RNP (page 6).
We have added Tchelidze et al 2019 in the references (page 11).
Indeed the DNA highlighting by osmium ammine could have been used in this work. However, this technique does not allow to visualize the nuclear compartments simultaneously. We preferred to use the TdT immunocytological method which is known for its high specificity and sensitivity.
Reviewer 2 Report
The work by Thelen et al. describes transmission electron microscopy approaches suitable for in situ analysis of chromatin organisation in budding yeast Saccharomyces cerevisiae. Cytochemical and immunocytochemical methods were applied. Moreover, induction of chromatin rearrangement by the hyperosmotic treatment was used.
This manuscript is clearly written and in detail. The methodological data are well documented by nice micrographs and statistic evaluation and I do not have not any major reservations. Just for illustration, it is a pity that authors did not try any cryo-techniques. In this case, the contrast between condense and de-condense chromatin can be improved, but it still remains rather weak. However, I believe this work will interest many workers in the field of the structure and function analysis of cell nucleus.
Author Response
We agree with the reviewer's comment about cryotechnology. Unfortunately, as the reviewer himself acknowledges, the contrast with cryotechnology is still rather weak. We preferred to opt for cytochemical and immunocytological techniques.